# Use of Lateral Flow Immunoassay to Characterize SARS-CoV-2 RBD-Specific Antibodies and Their Ability to React with the UK, SA and BR P.1 Variant RBDs

**DOI:** 10.3390/diagnostics11071190

**Published:** 2021-06-30

**Authors:** Enqing Tan, Erica Frew, Jeff Cooper, John Humphrey, Matthew Holden, Amanda Restell Mand, Jun Li, Shaya Anderson, Ming Bi, Julia Hatler, Anthony Person, Frank Mortari, Kevin Gould, Shelly Barry

**Affiliations:** 1Diagnostic Reagents Division, Bio-Techne Corporation, Devens Site, Devens, MA 01434, USA; Enqing.Tan@bio-techne.com (E.T.); Erica.Frew@bio-techne.com (E.F.); Matthew.Holden@bio-techne.com (M.H.); Amanda.Mand@bio-techne.com (A.R.M.); 2Reagent Solutions Division, Bio-Techne Corporation, Minneapolis Site, Minneapolis, MN 55413, USA; Jeff.Cooper@bio-techne.com (J.C.); John.Humphrey@bio-techne.com (J.H.); Jun.Li@bio-techne.com (J.L.); Shaya.Anderson@bio-techne.com (S.A.); Ming.Bi@bio-techne.com (M.B.); Julia.Hatler@bio-techne.com (J.H.); Anthony.Person@bio-techne.com (A.P.); 3Corporate Development, Bio-Techne Corporation, Minneapolis Site, Minneapolis, MN 55413, USA; Frank.Mortari@bio-techne.com; 4Diagnostic Reagents Division, Bio-Techne Corporation, San Marcos Site, San Marcos, CA 92078, USA

**Keywords:** COVID-19, SARS-CoV-2 variant, lateral flow immunoassay, spike protein, receptor binding domain (RBD), neutralizing antibody, therapeutic antibody cocktail, epitope binning, rapid neutralization test, ACE2

## Abstract

Identifying anti-spike antibodies that exhibit strong neutralizing activity against current dominant circulating variants, and antibodies that are escaped by these variants, has important implications in the development of therapeutic and diagnostic solutions and in improving understanding of the humoral response to severe acute respiratory syndrome coronavirus 2 (SARS-CoV-2) infection. We characterized seven anti-SARS-CoV-2 receptor binding domain (RBD) antibodies for binding activity, pairing capability, and neutralization activity to SARS-CoV-2 and three variant RBDs via lateral flow immunoassays. The results allowed us to group these antibodies into three distinct epitope bins. Our studies showed that two antibodies had broadly potent neutralizing activity against SARS-CoV-2 and these variant RBDs and that one antibody did not neutralize the South African (SA) and Brazilian P.1 (BR P.1) RBDs. The antibody escaped by the SA and BR P.1 RBDs retained binding activity to SA and BR P.1 RBDs but was unable to induce neutralization. We demonstrated that lateral flow immunoassay could be a rapid and effective tool for antibody characterization, including epitope classification and antibody neutralization kinetics. The potential contributions of the mutations (N501Y, E484K, and K417N/T) contained in these variants’ RBDs to the antibody pairing capability, neutralization activity, and therapeutic antibody targeting strategy are discussed.

## 1. Introduction

The continued emergence of SARS-CoV-2 variants has raised concerns and challenges for the control, prevention, and management of the coronavirus disease (COVID-19) [1]. Currently, the circulating variants of greatest concern include the United Kingdom variant (B.1.1.7 lineage, UK) [2,3], the SA variant (B.1.351 lineage) [4], the BR P.1 (B.1.1.28.1 lineage) [5], the Brazilian variant P.2 (B.1.1.28.2 lineage, BR P.2) [6], the Denmark mink variant (B.1.1.298 lineage, DM) [7], the California variants (B.1.429/427 lineage, CA) [8], the New York variants (B.1.526/525 lineage, NY) [9], and the more recent Indian variant (B.1.617 lineage, IN) [10]. In the absence of an effective strategy to curb the spread of SARS-CoV-2 virus infection, and due to more individuals harboring the virus, it is inevitable that more variants are likely to emerge.

The SARS-CoV-2 virus infects mammalian cells by attaching transmembrane spike proteins (S protein) to the angiotensin-converting enzyme 2 receptors (ACE2) found on the surface of the human target cells [11,12]. Hence, inhibiting the binding of SARS-CoV-2 spike protein to ACE2 has been the primary strategy behind most SARS-CoV-2 vaccines [13,14], therapeutic antibodies [15,16], and therapeutic soluble ACE2 molecules [17]. It is evident that the receptor binding domain (RBD) of the viral spike protein plays a critical role in the binding of SARS-CoV-2 to ACE2.

As of 14 May 2021, there are two therapeutic neutralizing antibody cocktails in use for the treatment of COVID-19 patients that have received emergency use authorization from the FDA [15,16]. Regeneron’s REGN-COV2 is a combination of two anti-SARS-CoV-2 RBD monoclonal antibodies (REGN10933 and REGN10987), and Eli Lily’s cocktail is a combination of two anti-SARS-CoV-2 RBD monoclonal antibodies (LY-CoV555 and LY-CoV016). The emergence of SARS-CoV-2 variants has unfortunately caused monoclonal antibody therapies and spike protein-based vaccines to be less effective than was determined in the original clinical studies [18,19]. In fact, due to the sustained increase in COVID-19 viral variants that are resistant to the LY-CoV555 antibody, the FDA recently revoked the emergency use authorization for this monoclonal antibody (LY-CoV555) monotherapy [20].

SARS-CoV-2 variants contain numerous mutations or deletions along the entire viral spike protein, but this report focuses on the key mutations in the RBD that have a direct impact on RBD-ACE2 interaction and the escape mechanism of the virus from neutralizing antibodies. Figure 1 summarizes currently circulating SARS-CoV-2 variants and their respective mutations within the spike RBD, which include the following: N501Y in the UK, SA, and BR-P.1 variants [2,3]; E484K/Q in the SA, BR P.1, BR P.2, NY, and IN variants [4,5,6,9,10]; K417N/T in the SA and BR P.1 variants [5,6]; L452R in the CA and IN variants [8,10]; S477N in some NY variants [9]; and Y453F in the Denmark mink variant [7]. These mutations cause a higher rate of viral infectivity, enhanced disease severity, and escape of the antibody’s neutralization action, resulting in reduced vaccine efficacy [2,4,7,18]. By completely mapping the SARS-CoV-2 RBD mutations, Starr et al. was able to demonstrate that the RBD containing the E484K mutation escapes the LY-CoV555 antibody, while the RBD containing the K417N/T mutations escapes the LY-CoV016 antibody [21].

To better understand how mutations mediate escape from an antibody’s neutralizing activity and to identify anti-RBD antibodies for potential diagnostic and therapeutic uses against SARS-CoV-2 variant infection, we employed a straightforward lateral flow immunoassay to characterize seven anti-RBD monoclonal antibodies for their binding activity, immunoassay pairing capability, and neutralizing activity toward SARS-CoV-2 RBD and the UK, SA, and BR P.1 variant RBDs. The objectives were three-fold: (1) to screen and identify variant-specific antibodies or escaped neutralizing antibodies for potential diagnostic applications, (2) to characterize and identify broadly potent neutralizing antibodies against SARS-CoV-2 and the variant RBDs for improved neutralization strategies, and (3) to explore the use of a rapid lateral flow-based dipstick assay and lateral flow cassette assay (LFA) for such studies. Using this simple LFA assay, we report our findings that two of the seven antibodies studied showed broad, potent neutralizing activity against all four RBDs. One antibody had strong neutralizing activity against the SARS-CoV-2 and UK variant RBDs but was unable to neutralize the SA and BR P.1 variant RBDs, and a combination of two antibodies from different epitope bins produced an additive effect in the neutralization activity against all four RBDs.

## 2. Materials and Methods

### 2.1. Materials

#### 2.1.1. Anti-SARS-CoV-2 Spike RBD Antibodies

Seven anti-SARS-CoV-2 spike RBD murine monoclonal antibodies (Ab1–Ab7) were selected for this study. Ab1 (clone No. 1035709), Ab2 (clone No. 1035740), Ab3 (clone No. 1035753), and Ab4 (clone No. 1035762) were generated using a Spodoptera frugiperda, Sf 21 (baculovirus) derived SARS-CoV-2 S1 subunit as the immunogen. Ab5 (clone No. 1035419) was generated using human embryonic kidney (HEK) cell, HEK293-derived SARS-CoV-2 spike RBD (R319-F541) protein as the immunogen. Ab6 (clone No. 1035224) and Ab7 (clone No. 1035240) were generated using the SARS-CoV-2 S1 subunit as the immunogen in a separate fusion. The screening and selection of these anti-RBD monoclonal antibodies during the hybridoma process was performed using an antigen down ELISA (enzyme-linked immunosorbent assay) assay (i.e., using a microtiter plate coated with SARS-CoV-2 spike RBD protein) to select strong binders. Two anti-nucleocapsid protein monoclonal antibodies (clone No. 1035101 and clone No. 1035138), generated using SARS-CoV-2 nucleocapsid full-length protein as the immunogen, were used as negative controls. All antibodies used were produced by Bio-Techne Corporation (Minneapolis, MN, USA).

#### 2.1.2. Recombinant SARS-CoV-2 and Variant RBDs

All recombinant RBD proteins were generated by Bio-Techne Corporation (Minneapolis, MN, USA), corresponding to NCBI (National Center for Biotechnology Information) reference sequence accession number YP_009724390.1 [22], using the HEK293 expression system. These proteins included the SARS-CoV-2 RBD (R319-F541), the UK variant RBD (R319-F541 with N501Y), the SA variant RBD (R319-F541 with K417N, E484K, and N501Y), and the BR P.1 variant RBD (R319-F541 with K417T, E484K, and N501Y). The secreted recombinant proteins were purified from the conditioned media by nickel chelating chromatography, followed by size exclusion chromatography. All the recombinant RBD constructs included a C-terminal 6-His tag.

#### 2.1.3. Other Materials

Triton X-100, 30% bovine serum albumin (BSA) solution, 30% Brij-35 solution, 10× phosphate buffered saline (PBS), and other chemicals were purchased from Millipore-Sigma (Burlington, MA, USA). Casein solution (1%, *w*/*v*) in a Tris buffer with a pH of 7.4 was purchased from Thermo Fisher Scientific (Waltham, MA, USA). Recombinant human ACE2 protein, goat polyclonal anti-chicken IgY antibody, and chicken IgY protein (cIgY) were acquired from Bio-Techne Corporation (Minneapolis, MN, USA).

### 2.2. Lateral Flow Dipstick, Neutralization Test Devices, and Lateral Flow Immunoasay

#### 2.2.1. Preparation of Antibody-AuNP Conjugate

The anti-RBD and anti-nucleocapsid antibodies were coated onto 40-nm citrate-protected gold nanoparticles (AuNP, nanoComposix, San Diego, CA, USA) using a modified procedure from the gold nanoparticle manufacturer’s suggested protocol. Briefly, an AuNP solution of 20 OD (optical density) was combined with high purity water at a 1:4 volume/volume ratio (1 part of a 20 OD AuNP solution in 0.02 mM sodium citrate and 4 parts of water). An antibody suspended in 1× PBS was added to the AuNP solution at 5% or less of the total reaction volume, with the antibody-to-AuNP ratio of 50 μg of antibody per mL of 20 OD AuNP. Due to partial particle aggregation during the adsorption process, the amount of Ab7 was reduced to 20 μg per mL of the 20 OD AuNP solution. For the discriminative binding study, Ab4 and Ab5 were coated at a rate of 20 μg per mL of 20 OD AuNP solution. After the AuNP and antibody mixture was incubated at ambient room temperature for 30 min, BSA was added to a final concentration of 5 mg/mL to block the remaining AuNP surface reactivity. After another 30-min incubation period, the AuNP and antibody conjugate solution was centrifuged at 3800 RCF (relative centrifugal force) for 10 min to pellet the antibody-AuNP (Ab-AuNP) conjugate. The conjugate was washed twice with a wash and storage buffer and then finally resuspended in the wash and storage buffer and stored at 4 °C until use. The optical density of the conjugate solution was confirmed by an absorbance reading at 525 nm using a NanoDrop 2000 spectrophotometer. The wash and storage buffer was composed of 0.05× PBS with a pH of 7.4, containing 0.5% (*w*/*v*) BSA and 0.05% (*w*/*v*) sodium azide.

#### 2.2.2. Lateral Flow Dipstick and Assay Procedure for Antibody Pairing Capability and Epitope Binning

Each dipstick consisted of a polystyrene card backing with a 25-mm nitrocellulose membrane and a 17-mm absorbent pad. The dipsticks were prepared using a 60 mm × 300 mm FF120HP Whatman membrane card (Cytiva, Marlborough, MA, USA). The 20-mm wide adhesive portion was removed using a paper cutter, and then a 17 mm × 300 mm cellulose fiber sample pad strip (Millipore-Sigma, Burlington MA, USA) was attached to the 15-mm wide adhesive portion as the absorbent pad, with ~2 mm overlapping the nitrocellulose membrane. The card assembly was cut into 40-mm wide sections (dipsticks) using a Matrix 2360 Programmable Shear.

The dipsticks were then spotted with a nine-spot protein array: seven spots for the anti-RBD antibodies and two spots for the negative controls (i.e., the anti-NP antibody and BSA). Each protein was diluted to 1 mg/mL in 1× PBS with a pH of 7.4 and manually pipetted onto the nitrocellulose membrane at 1 μL per spot. The coated dipsticks were dried in a 37 °C oven with circulating air for a minimum of 30 min prior to use; longer term storage occurred in a plastic bag with desiccant.

Immediately prior to running an assay for the epitope binning and pairing capability study, a 300-μL sample mixture was prepared by mixing three solutions: 150 μL RBD (1 μg/mL) in an assay run buffer, 80 μL Ab-AuNP conjugate (4 OD) in a wash and storage buffer, and 70 μL of an assay run buffer. To start the assay, a spotted dipstick was placed in a reservoir with the nitrocellulose end at the bottom, and then the freshly prepared sample mixture was transferred into the reservoir, allowing the sample mixture to flow up the nitrocellulose membrane to the absorbent pad. After the sample mixture ran for 12–15 min, an additional 250 μL of the assay run buffer was added to rinse off any unbound material for another 12–15 min. Thus, each assay run took approximately 30 min. For the antibodies with good pairing capability, RBD protein would bind to the Ab-AuNP detector and then be captured by the spotted antibody to form a red-colored crescent line or circular spot (i.e., a sandwich assay). The red color intensity generated by the accumulated 40-nm gold nanoparticles at the coated antibody spots was stable for approximately 90 min. After that, the dipstick dried out, and the red color became lighter and permanent. For consistency, all assay images of the lateral flow spot dipsticks for antibody binning and binding characteristics were taken within 30 min after the assay was completed. For the antibodies with poor pairing capability, the RBD-Ab-AuNP complex would not be captured by the spotted antibody, producing little to no color change. The assay run buffer consisted of 1× PBS with a pH of 7.4, with 1.5% (*w*/*v*) BSA, 0.25% (*w*/*v*) Tween-20, 0.2% (*w*/*v*) casein, and 0.025% (*w*/*v*) sodium azide.

The assay procedure for evaluating the comparative binding characteristics to four RBD proteins was further modified for better assay sensitivity and specificity. All steps were the same as the above procedure for the epitope binning and pairing capability study, except that the 300-μL sample mixture was prepared by mixing three solutions: 50 μL RBD in a Brij-35 assay buffer, 40 μL of Ab-AuNP conjugate (4 OD) in a wash and storage buffer, and 210 μL of the Brij-35 assay buffer. The Brij-35 assay buffer (250 μL per dipstick) was also used as the assay run buffer. The Brij-35 assay buffer was composed of 1× PBS with a pH of 7.4, containing 1% (*w*/*v*) BSA, 0.1% (*v/v*) TX-100, 0.3% (*v*/*v*) Brij-35, 0.3% (*w*/*v*) casein, and 0.05% (*w*/*v*) sodium azide.

At the end of an assay, each dipstick was imaged using an iPhone camera to document the assay results.

#### 2.2.3. Lateral Flow Cassette and Assay Procedure for Neutralization Kinetics

Lateral flow neutralization test devices were developed and manufactured according to the procedures developed in our laboratory. Briefly, recombinant human ACE2 was striped in the “test zone”, and the goat polyclonal anti-chicken IgY antibody was striped in the “control zone” as the capture agent using an IsoFlow Reagent Dispenser. Recombinant RBD and cIgY were conjugated to the 40-nm gold nanoparticles as the detectors for the test zone and the control zone, respectively. The AuNP-RBD and AuNP-cIgY conjugates were combined in a drying down buffer containing salt, stabilizer, and AuNP releasing agents and sprayed onto conjugate pad strips using an IsoFlow Reagent Dispenser. The sprayed conjugate pad strips were dried in a 37 °C oven and then stored in a sealed foil pouch with desiccants until use. By alternating the AuNP-RBD conjugate while keeping other components the same, four types of rapid neutralizing antibody test cassettes were prepared for the SARS-CoV-2 RBD, the UK variant RBD, the SA variant RBD, and the BR P.1 variant RBD.

The neutralization antibody test strip consisted of a plastic backing card attached with a sample pad, a conjugate pad with dried gold conjugate detectors, a nitrocellulose membrane striped with the capture antibodies, and an absorbent (wicking) pad. Each test strip was assembled into a plastic cassette, sealed in a foil pouch with desiccant, and stored at ambient room temperature. The dropper bottle used for introducing the neutralization assay buffer had a drop size of ~25 μL per drop. The neutralization assay buffer consisted of 1× PBS with a pH of 7.4, containing 3% (*w*/*v*) BSA, 0.5% (*v*/*v*) Tween-20, and 0.05% (*w*/*v*) sodium azide.

A neutralization kinetic curve was generated for all seven antibodies at different concentrations to determine the percentage of neutralization against all four RBDs. Each antibody was diluted in a neutralization assay buffer with antibody concentrations at 10, 2, 0.5, and 0.1 μg/mL and a zero-antibody control. The assays were run in stacking modes, with each assay delaying for 30 s for up to 20 tests per run or delaying for 60 s for up to 10 tests per run. Each run took slightly more than 20 min, including 10 min of assay time and 10 min for reading all the test cassettes of the same run using an RDS-2500 LFA reader. For each cassette assay, an aliquot of 20 μL of a diluted antibody solution was added to the sample port of a test cassette to start the assay. After the antibody sample absorbed for ~25 s, three drops (~75 μL) of a neutralization assay buffer was added to the sample port using a dropper bottle. The signal intensities of the test zone, the control zone, and a reference negative zone were measured at 10 min from the start of the assay using an RDS-2500 LFA reader. If the antibody had no neutralizing activity, the AuNP-RBD conjugate was captured by immobilized ACE2 at the test zone, and a red test line was formed. If the antibody had neutralizing activity, then the antibody would bind to the AuNP-RBD and prevent it from being captured by immobilized ACE2, resulting in little to no signal. The cIgY control zone served to verify (1) the biological reagents of the test strip were active, (2) the sample mixture and assay run buffer flowed through the detection membrane properly, and (3) the performance of the neutralization assay buffer. The reference negative zone, depicting the background of the detection membrane after the sample and assay run buffer flow through the cassette, served as an image quality control.

Each test would have three intensity values: one each for the control zone, the test zone, and the reference negative zone. For comparison of the neutralizing activity against SARS-CoV-2 and three variant RBDs, four sets of data were collected for each antibody. Two or three replicates were run for each antibody concentration except the zero-antibody control, with 6–8 replicates per test device type.

### 2.3. Calculation of the NC_50_ Value for Each Neutralization Kinetic Curve

The NC_50_ in μg/mL is the concentration of antibody that yields a 50% inhibition or neutralization of the maximal RBD-ACE2 binding capacity for a given neutralization cassette type. The intensity value of the test zone was used to determine the NC_50_ value.

To calculate the NC_50_ value per device type for each antibody, the mean signal intensity of the test zone at the zero-antibody concentration was calculated to represent the maximum RBD-ACE2 binding activity (i.e., 0% neutralization). When an antibody has neutralization activity, the binding capacity of the AuNP-RBD to immobilized ACE2 is reduced. The difference between the observed binding activity of each test and the maximum binding activity is the neutralization activity, which is then converted to an individual percentage of neutralization (%Neutralization). The %Neutralization of each individual test was calculated using the conversion formula shown below:%Neutralization of individual test =
(Mean intensity _zero-Ab_ − Signal Intensity _individual test_)/Mean Intensity _zero-Ab_.

The mean %Neutralization was calculated from the converted %Neutralization replicate values for each test condition. A scatterplot was generated using the antibody concentration as the *x*-axis and the mean %Neutralization value as the *y*-axis. A semi-logarithmic curve fitting was performed for the determination of the NC_50_ value of each antibody against all RBDs (Figure 2).

### 2.4. Instrumentation and Statistics

The NanoDrop 2000 spectrophotometer was purchased from Thermo Fisher Scientific (Waltham, MA, USA). The RDS-2500 LFA reader was acquired from Detekt Biomedical LLC (Austin, TX, USA) with the default R/G/B of 0/1/0 settings. The Matrix 2360 Programmable Shear was purchased from Kinematic Automation (Sonora, CA, USA). The IsoFlow reagent dispenser was purchased from Imagene Technology (Lebanon, NH, USA). All calculations were carried out using Microsoft Excel.

## 3. Results

### 3.1. Antibody Pairing Capability and Epitope Binning Using SARS-CoV-2 RBD

Seven anti-SARS-CoV-2 RBD antibodies were studied for their pairing capability and epitope binning using a lateral flow dipstick immunoassay. These antibodies were spotted onto lateral flow nitrocellulose membranes as capture agents and then reacted with SARS-CoV-2 RBD in the presence of one Ab-AuNP conjugate as a detection agent. A non-competing antibody would pair and form a sandwich immunocomplex with the SARS-CoV-2 RBD, appearing as a dark red colored crescent line or spot on the nitrocellulose membrane. A competing antibody will not form a sandwich immunocomplex, appearing as a light red colored spot or no spot at all. The sandwich immunoassay principle is shown in Figure 3A. The immunodetection results for each detector anti-RBD Ab-AuNP conjugate and the negative control anti-NP Ab-AuNP conjugate are presented in Figure 3B–I. Based on the binding pattern and pairing capability, these antibodies were grouped into three distinct epitope bins: bin A (Ab1 and Ab4), bin B (Ab2 and Ab5), and bin C (Ab3, Ab6, and Ab7) (Table 1). The fact that these seven monoclonal antibodies were generated through three distinct fusions, along with the classification of these antibodies into three epitope bins, suggests that these epitopes constitute dominant antigenic domains of SARS-CoV-2 RBD and play a very important role in the natural immune response to SARS-CoV-2 infection and, presumably, vaccination.

### 3.2. Comparative Binding Characteristics to Three Variant RBDs vs. SARS-CoV-2 RBD

To investigate whether the above antibody pairings could detect the UK, SA, and BR P.1 variant RBDs, similar dipstick sandwich immunoassays were performed using the same capture antibody panel along with three representative epitope bin detectors: Ab4-AuNP, Ab5-AuNP, and Ab6-AuNP conjugates. Our initial experiment showed that the Ab7-Ab4 (capture/detector) pair discriminatively detected the four RBDs. Specifically, the SARS-CoV-2 and UK RBDs were strongly detected, and the SA and BR P.1 RBDs were weakly detected by the Ab7-Ab4-AuNP pair. All other pairing options that were tested did not show discriminative detection of these four RBDs.

To confirm these discriminative binding characteristics, the dipstick immunoassays were carried out using serially diluted RBD protein solutions, ranging from 0.001 μg/mL to 1 μg/mL. Since Ab6 and Ab7 belong to the same epitope bin, only the Ab4-AuNP and Ab5-AuNP detectors were used in this confirmatory study. As shown in Figure 4A, with the Ab5-AuNP as the detector and the Ab1, Ab3, Ab4, Ab6, and Ab7 antibodies as the capture agents, all pairing options strongly detected all four RBDs with a detection sensitivity of 0.001 μg/mL. Similarly, with the Ab4-AuNP as the detector (Figure 4B) and the antibodies Ab2, Ab3, Ab5, and Ab6 as the capture agents, all pairing options detected all four RBDs with a detection sensitivity of 0.01 μg/mL. Overall, Ab5-AuNP appeared to have a better detection sensitivity than Ab4-AuNP as the detector.

Interestingly, when the Ab4-AuNP detector was paired with Ab7 as the capture agent, the SARS-CoV-2 and UK RBDs were strongly detected with a detection sensitivity of 0.01 μg/mL. However, the SA and BR P.1 RBDs were weakly detected, with a detection sensitivity between 0.1 and 1.0 μg/mL. This discriminative binding behavior of the Ab7-Ab4 pair for the detection of SARS-CoV-2 RBD and the three variant RBDs was also observed when Ab7 was paired with Ab1 (another bin A epitope antibody). Given that the Ab5-AuNP pairing with Ab7 capture strongly detected the SA and BR P.1 RBDs, and that the Ab4-AuNP pairing with antibodies Ab2, Ab3, Ab5, and Ab6 also strongly detected the SA and BR P.1 RBDs, both Ab7 and Ab4 appeared to bind to the SA and BR P.1 RBDs well alone. Thus, the weak binding activity of the Ab7-Ab4 pair (and Ab7-Ab1 pair) to the SA and BR P.1 RBDs compared with the SARS-CoV-2 and UK RBDs indicates that the E484K and K417N/T mutations contained in the SA and BR P.1 RBDs most likely induced conformational changes near or within the epitopes of where these antibodies bind. The conformational change could result in steric interference between Ab7 and the bin A epitope antibodies.

### 3.3. Neutralizing Kinetics of Individual Antibody and Combination of Antibodies

To determine the neutralization kinetics of these antibodies against the three variant RBDs in comparison to the SARS-CoV-2 RBD, lateral flow neutralization cassette assays were carried out using rapid neutralizing antibody test devices made for the SARS-CoV-2, UK, SA, and BR P.1 RBDs, respectively. The lateral flow neutralization test principle is illustrated in Figure 5A. Essentially, AuNP-RBD conjugates were captured by immobilized ACE2 protein in the absence of a neutralizing antibody, forming a red-colored line at the test zone, but if a neutralizing antibody was present, then it would bind to the AuNP-RBD and prevent the AuNP-RBD from being captured by immobilized ACE2 protein. The signal intensity of the test zone was inversely correlated with the concentration of the neutralizing antibody. The signal intensity of the control zone was not affected by the neutralizing antibody. A representative set of lateral flow neutralizing antibody test cassettes is depicted in Figure 5B.

The neutralization kinetics of each individual antibody against the SARS-CoV-2 RBD and three variant RBDs was studied using serially diluted antibody solutions targeting antibody concentrations at 10, 2, 0.5, 0.1, and 0 μg/mL suspended in a neutralization assay buffer (Figure 6A–G). The NC_50_ value was determined from the signal intensity of the test zone as described in the method section (Figure 2B). Five antibodies (Ab1, Ab2, Ab4, Ab6, and Ab7) showed strong neutralizing activity against the SARS-CoV-2 RBD, with NC_50_ values between 0.34 and 1.43 μg/mL, and two antibodies (Ab3 and Ab5) showed a moderate neutralizing activity against the SARS-CoV-2 RBD with an NC_50_ value of 1.66 and 3.27 μg/mL, respectively, indicating that the epitopes of these seven antibodies were all near or in the RBD interface zone. The NC_50_ values for the seven antibodies against all four RBDs are summarized in Table 2.

With regard to the UK, SA, and BR P.1 variant RBDs, two antibodies, Ab1 and Ab4 of bin A, maintained strong neutralizing activity against all three variant RBDs, with NC_50_ values between 0.69 and 1.08 μg/mL, suggesting that these two antibodies targeted an epitope that was not affected by the mutations (N501Y, E484K, and K417N/T) contained in these three variant RBDs. The remaining five monoclonal antibodies either partially or completely lost their neutralizing activity against at least one variant RBD. Antibody Ab7 showed strong neutralizing activity against the SARS-CoV-2 and UK RBDs, with an NC_50_ value of 0.34 and 0.86 μg/mL, respectively, but completely lost its neutralizing activity against the SA and BR P.1 RBDs, with an NC_50_ value greater than 45 μg/mL (not able to be precisely calculated due to the flat curve shape). This indicates that the Ab7 epitope was not affected by the shared N501Y mutation but was very likely affected by the E484K or K417N/T mutations. Antibody Ab5 showed reduced neutralizing activity against the SARS-CoV-2 RBD, with an NC_50_ value of 3.27 μg/mL. This reduction in neutralizing activity was even greater for the three variant RBDs, with NC_50_ values between 24 and 45 μg/mL. This suggests that the Ab5 epitope was in the proximity of the N501Y mutation, which is the only mutation shared among the UK, SA, and BR P.1 RBDs.

We then evaluated the neutralization activity of an antibody cocktail, combining Ab1 of the bin A epitope and Ab2 of the bin B epitope (Figure 6H). We observed a partial additive effect in the neutralizing activity of this antibody cocktail against the SARS-CoV-2 RBD and all three variant RBDs, yielding NC_50_ values between 0.46 and 0.82 μg/mL, which were all less than their corresponding NC_50_ values of Ab1 (between 0.63 and 1.08 μg/mL) or Ab2 (between 1.35 and 1.83 μg/mL) alone. These results indicate that a combination of two or three antibodies of different epitope bins could be used to enhance the neutralization capabilities of a therapeutic cocktail and that such an enhancement can be readily characterized by this lateral flow neutralization cassette assay.

### 3.4. Analysis and Functional Epitope Arrangement Map

Based on the binding characteristics and the neutralization activity against the SARS-CoV-2 RBD and all three variant RBDs, a functional arrangement map of these seven antibodies versus each RBD was generated. For the sandwich immunodetection of the SARS-CoV-2 RBD, these antibodies paired freely among the three epitope bins (Figure 7A). Five antibodies (Ab1, Ab2, Ab4, Ab6, and Ab7) showed strong neutralizing activity, with an NC_50_ value less than 1.5 μg/mL, and two antibodies (Ab3 and Ab5) showed moderate neutralizing activity, with an NC_50_ value between 1.5 and 15 μg/mL (Table 2). Overall, these seven antibodies paired freely among the three epitope bins and showed moderate to strong neutralizing activities against the SARS-CoV-2 RBD.

Similar to the SARS-CoV-2 RBD, these antibodies paired freely among the three epitope bins for sandwich immunodetection of the UK RBD (Figure 7B). However, their neutralizing activity differed significantly. Four antibodies (Ab1, Ab2, Ab4, and Ab7) showed strong neutralizing activity against the UK RBD, with NC_50_ values less than 1.5 μg/mL, while two antibodies (Ab3 and Ab6) had moderate neutralizing activity, with NC_50_ values between 1.5 and 15 μg/mL, and one antibody (Ab5) had weak neutralizing activity, with an NC_50_ value of 27.63 μg/mL (Table 2). The reduced neutralizing activity of Ab5 against the UK RBD was likely caused by the N501Y mutation, the only reported mutation contained in the UK variant RBD.

The SA and BR P.1 RBDs showed similar binding and neutralizing characteristics to these seven antibodies. For sandwich immunodetection of these two variant RBDs, these antibodies could pair freely among the three epitope bins, except the pairing of Ab7 (capture) and bin A epitope antibodies as a detector. Unlike the SARS-CoV-2 and UK RBDs, when antibody Ab7 was used as a capture agent and then paired with Ab1 or Ab4 of the bin A epitope, the detection of both SA RBD and BR P.1 RBD in the sandwich assay format was significantly reduced (Figure 7C,D). Five antibodies (Ab1, Ab2, Ab3, Ab4, and Ab6) had either strong or moderate neutralizing activity against these two RBDs, with NC_50_ values less than 15 μg/mL (Table 2). Consistent with the reduced neutralizing activity for the UK RBD, antibody Ab5 had weak neutralizing activity against the SA and BR P.1 RBDs, with NC_50_ values greater than 15 μg/mL. Furthermore, antibody Ab7 did not show any neutralizing activity against the SA and BR P.1 RBDs at the concentration range tested. The E484K or K417N/T mutations contained in the SA and BR P.1 RBDs very likely influenced Ab7 to lose its neutralizing activity.

## 4. Discussion

SARS-CoV-2 spike protein is a critical component for SARS-CoV-2 to adhere to and enter mammalian cells [11,12]. The spike protein and its RBD are highly antigenic and have been the primary target of numerous recently developed vaccines and therapeutics [13,14,15,16]. Given that the seven antibodies evaluated in this report were generated from three distinct fusions, the classification of these antibodies into three epitope bins by virtue of their binding and neutralization characteristics suggests that their corresponding target regions on the RBD protein belong to dominant antigenic epitopes, which could elicit protective humoral immune responses to SARS-CoV-2 infection or vaccination. The two antibodies of bin A epitopes Ab1 (clone 1035709) and Ab4 (clone 1035762) exhibited broad and potent neutralization activity against all four RBDs and offered useful insight into the development of therapeutic antibodies for emerging virus variants. While more studies are required to more definitively identify the epitopes we revealed, our studies suggest that therapeutic antibodies targeting epitopes similar to bin A are likely to provide better protection against the UK, SA, and BR P.1 variants, while therapeutic antibodies targeting epitopes similar to bin B and bin C are more likely to be escaped by these variants. Based on the studies showing that three variants (UK, SA, and BR P.1 RBD mutants) escaped the therapeutic LY-CoV555 antibody and two variants (SA and BR P.1 RBD mutants) escaped the LY-CoV016 antibody, Starr et al. suggested that both the SA and the BR P.1 variants may also escape the antibody cocktail of LY-CoV555 and LY-CoV016 [21]. Thus, having rapid tools available to study the neutralization activities of antibodies, along with further elucidation of the exact nature of the epitopes recognized by our bin A antibodies (Ab1 and Ab4), could provide a useful framework for the rapid development of therapeutic cocktails against the constantly emerging SARS-CoV-2 variants.

Although numerous factors can affect the binding of SARS-CoV-2 spike protein to the ACE2 receptor [23], the virus spike protein RBD plays a central role in this interaction [11,12]. The N501Y RBD mutation, first found in the UK variant [3], has been demonstrated to exhibit stronger interaction force with the ACE2 receptor, which is associated with the increased infectivity of the UK variant [2,8,24]. With regard to the impact of N501Y mutation on the escape of a neutralizing antibody, Supasa et al. [25] reported that the UK variant is not neutralized as easily as SARS-CoV-2 by convalescent sera, vaccine sera, or some anti-RBD monoclonal antibodies, while Xie et al. [18] observed only a small reduction in neutralization activity against the UK variant by sera, elicited by two doses of the Pfizer vaccine BNT162b2. Our results demonstrate that Ab5 exerts significantly reduced neutralizing activity against the UK, SA, and BR P.1 variant RBDs, suggesting that the shared N501Y mutation not only increased ACE2-RBD interaction [2], but it also contributed to the escape phenomenon of the neutralizing antibody. Together, this may explain the weakened efficacy of vaccines or therapeutic antibodies against these variants.

As for the E484K mutation, characteristic of the SA and BR P.1 variant RBDs, it was reported to be “associated with escape from neutralizing antibodies”, which adversely affects the efficacy of spike protein-dependent COVID-19 vaccines [26]. Several studies have demonstrated that spike-targeted vaccines or convalescent plasma from human subjects with SARS-CoV-2 are less effective at neutralizing the SA variant [27]. In fact, Moderna Inc. recently initiated a clinical trial using a modified version of the spike RNA vaccine to counter the SA variant and perhaps other known variants [28]. Our results show that both the SA and BR P.1 variant RBDs completely escaped Ab7 neutralization, confirming that the E484/K417 mutations may be directly involved with the escape mechanism. Further characterization of the antibody Ab7 epitope could shed light on better understanding how to focus the humoral immune response on these challenging epitopes and avoid the escape mechanism of SARS-CoV-2. Additionally, having rapid test tools for clinical diagnostics and epidemiological studies for SARS-CoV-2 variant infection could prove useful in situations where timing to the development of effective clinical tools is mission critical.

Being able to monitor multiple parameters of potential antibody candidates is key to being able to make the right decisions on antibody selections for clinical utility. For example, the pairing of Ab7 of bin C with antibody Ab4 (and Ab1) of bin A showed very weak binding activity with the SA and BR P.1 RBDs in the sandwich immunoassay format. However, the Ab7 pairing with Ab5 and Ab2 of bin B did not show any apparent difference in the binding activities for all four RBDs studied. This discriminative binding behavior to the four RBDs suggests that the E484K or K417N/T mutations likely induced conformational changes near the epitopes, where these antibodies bind in the SA and BR P.1 variant RBDs. This in turn may have resulted in steric interference between Ab7 and the bin A antibodies, but no apparent steric interference between Ab7 and the bin B antibodies. Furthermore, such a mutation-induced conformational change of the spike molecule could contribute to the escape phenomenon of antibody Ab7 by the SA and BR P.1 variant RBDs. Since the neutralization activities of the bin A antibodies (Ab1 and Ab4) were not affected by the three mutations contained in the UK, SA, and BR P.1 variant RBDs, further investigation of the mechanism by which the E484K or K417N/T alters the binding and neutralizing activities of these antibodies may help in the identification of a better therapeutic epitope target.

Lateral flow assays are commonly used for rapid clinical testing, such as COVID-19 serological tests, antigen tests, certain molecular tests, and the neutralizing antibody test [29,30,31,32,33]. Wang et al. reported on the use of a lateral flow dipstick assay with wild-type and the South African spike S1 protein for the characterization of the neutralizing activity of post-vaccination plasma samples [29]. Our observation that the calculated NC_50_ values of five mouse monoclonal antibodies for the SARS-CoV-2 RBD were between 0.3 and 1.5 μg/mL appears to be consistent with a reported IC_50_ of 1.402 μg/mL for a mouse monoclonal antibody using a dipstick assay [29]. Our studies demonstrated that sandwich-based immunoassays, such as lateral flow assays, offer an attractive and cost-effective alternative in characterizing the antibody binding properties, epitope binning, and in vitro neutralizing kinetics of therapeutic antibodies and cocktails. Lake et al. and others have demonstrated that the lateral flow-based (surrogate) neutralizing antibody test performs similarly to the pseudo-virus or the authentic SARS-CoV-2 virus-based neutralizing antibody assays in assessment of the neutralizing antibody activity of convalescent samples against SARS-CoV-2 [29,30]. Therefore, our lateral flow-based rapid neutralizing antibody tests could potentially be used to assess the neutralizing antibody activities against the SARS-CoV-2 RBD and the UK, SA, and BR P.1 variant RBDs in human blood samples.

## 5. Patent

The work presented in this report was part of a provisional patent application filed at the USA patent office. No decision has been made at this point.

## Figures and Tables

**Figure 1 diagnostics-11-01190-f001:**
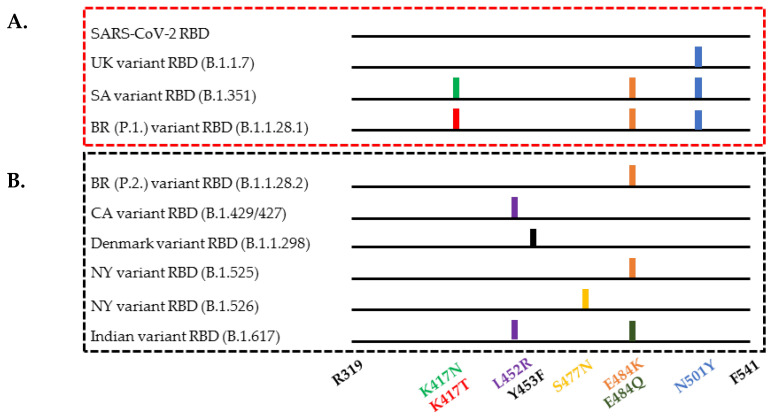
Mutations in the spike RBD protein of the currently circulating SARS-CoV-2 variants, compiled from these referenced articles [2,3,4,5,6,7,8,9,10,22]. (**A**) SARS-CoV-2 and three variant RBDs included in this study. (**B**) Variant RBDs not addressed in this study.

**Figure 2 diagnostics-11-01190-f002:**
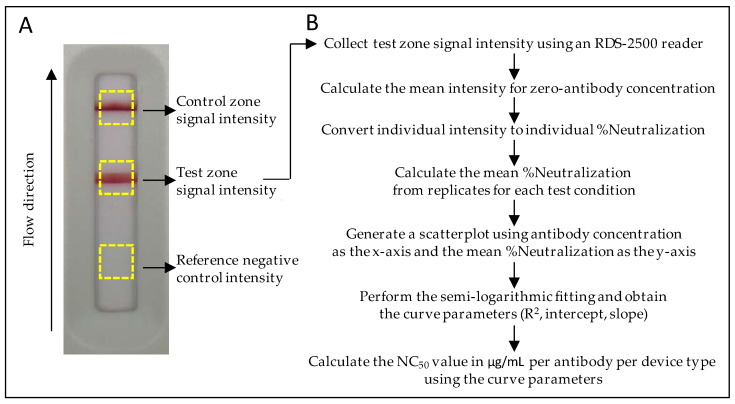
Collection of signal intensities and flowchart for the calculation of NC_50_ values from the test line signal intensity. (**A**) An example of a test cassette image recorded by the RDS-2500 LFA reader with designated zones for the collection of signal intensities. (**B**) Flowchart for the calculation of the NC_50_ value for each neutralization kinetic curve.

**Figure 3 diagnostics-11-01190-f003:**
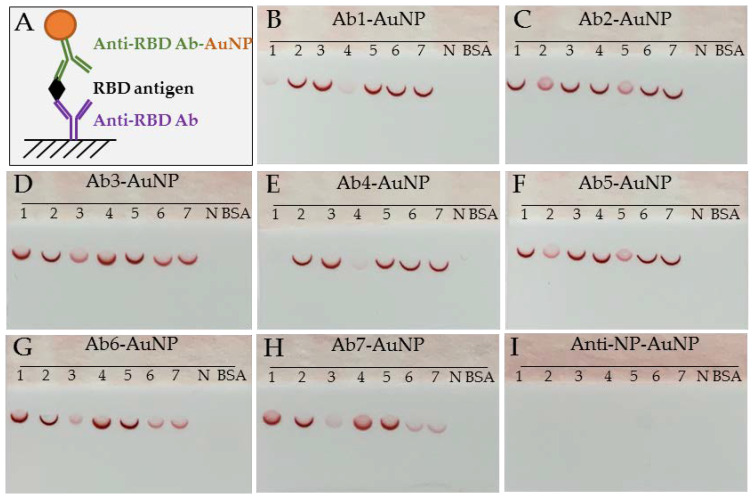
Antibody pairing capability and epitope binning. (**A**) Sandwich immunoassay principle. (**B**–**I**) Photographs of a representative set of lateral flow dipstick assays. Each dipstick has 9 spots: 7 spots for the antibodies of interest and 2 spots for the negative controls (i.e., anti-nucleocapsid antibody and BSA). (Note 1) A dark red colored crescent line or circular spot indicates strong binding activity, a light red colored crescent line or circular spot indicates weak binding activity, and an empty spot indicates no binding activity. (Note 2) All Ab-AuNP conjugates used were coated with 50 μg of antibody per mL of 20 OD AuNP solution, except Ab7-AuNP, which was coated with 20 μg of antibody per mL 20 OD AuNP. For anti-nucleocapsid antibodies, clone 1,035,138 (N) was used as a capture negative control, and clone 1,035,101 (anti-NP) was used as a detector negative control.

**Figure 4 diagnostics-11-01190-f004:**
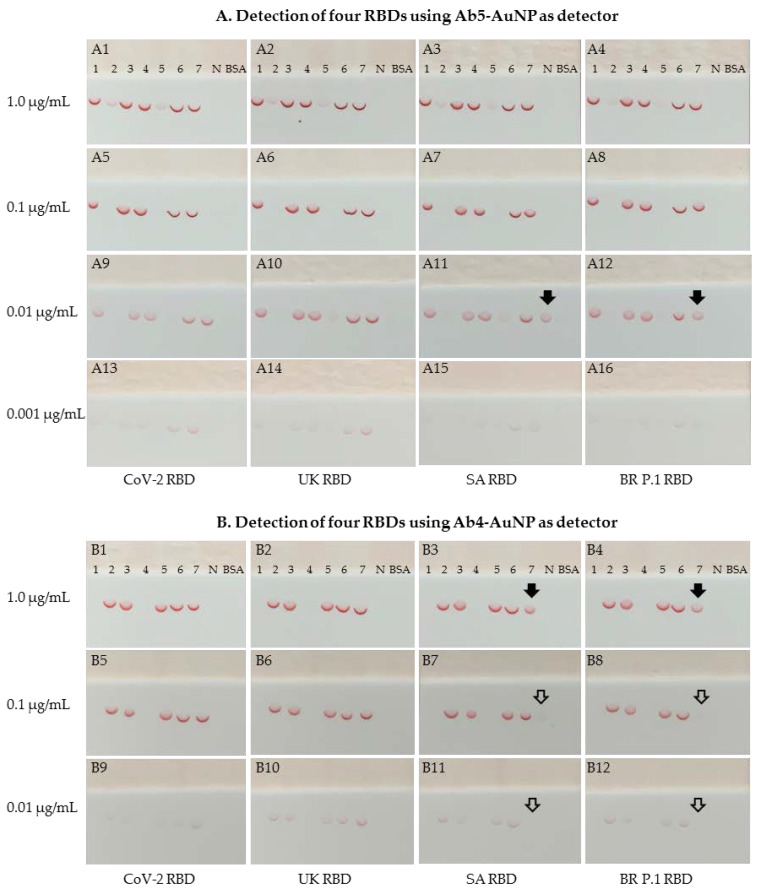
Comparative binding characteristics for four RBDs using Ab4-AuNP and Ab5-AuNP as detectors, with representative photographs of the test results. (**A**) Binding behavior of the Ab5-AuNP conjugate to four RBDs. (**B**) Binding behavior of the Ab4-AuNP conjugate to four RBDs. (Note 1) Solid arrows in the dipsticks (**A11**, **A12**, **B3**, and **B4**) indicate similar signal intensity at the Ab7 spot. Open arrows in the dipsticks (**B7**, **B8**, **B11**, and **B12**) indicate very faint or invisible signals at the Ab7 spot. (Note 2) Both the Ab4-AuNP and Ab5-AuNP conjugates were prepared with 20 μg of antibody per mL of 20 OD AuNP solution.

**Figure 5 diagnostics-11-01190-f005:**
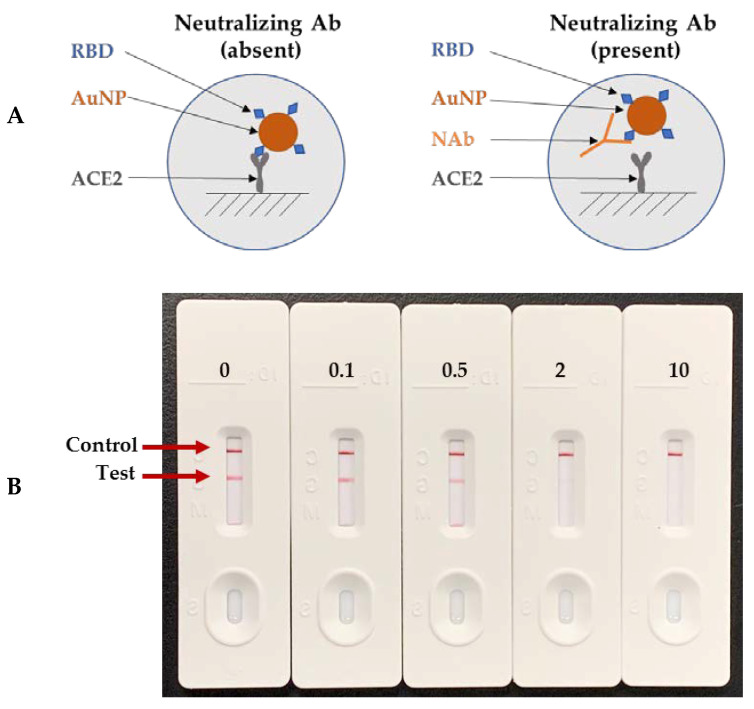
Neutralizing antibody test principle and images of the representative test devices. (**A**) Lateral flow cassette neutralization assay principle. (**B**) Photographs of a representative set of lateral flow neutralization test devices. (Units = neutralizing antibody concentration in μg/mL).

**Figure 6 diagnostics-11-01190-f006:**
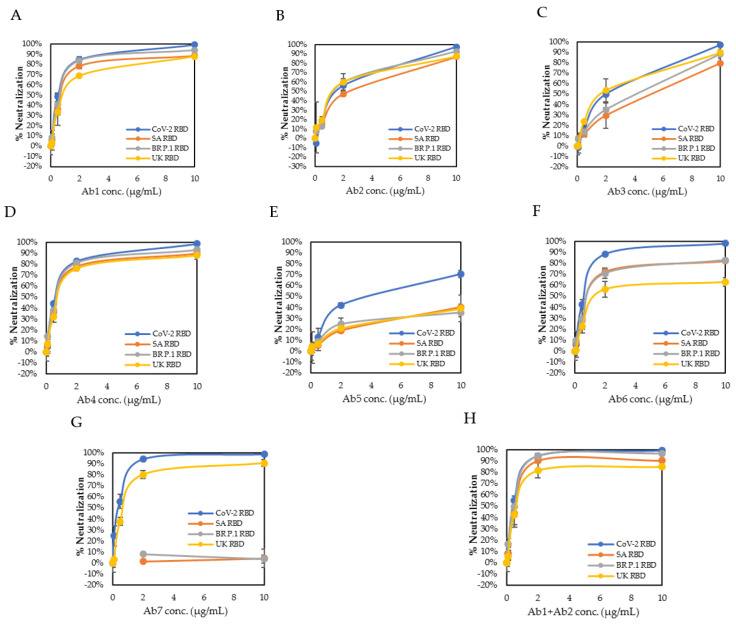
Neutralization kinetics by lateral flow cassette neutralization assays. (**A**–**G**) Kinetic curves for individual antibodies against the four RBDs. (**H**) Kinetic curve for the two combined antibodies against the four RBDs. Ab1 and Ab2 were used at the same concentration for each level.

**Figure 7 diagnostics-11-01190-f007:**
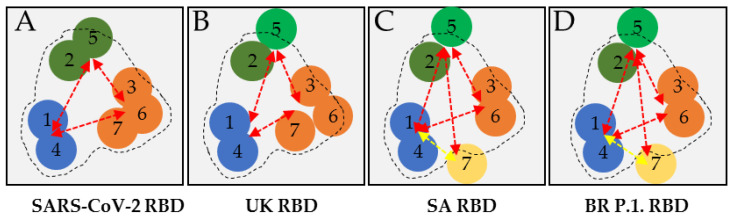
Functional arrangement map of the antibody pairing capability and neutralizing activity. (**A**–**D**) Reactivity arrangement map of seven monoclonal antibodies to SARS-CoV-2 RBD, UK RBD, SA RBD, and BR P.1 RBD, respectively. (Note 1) Blue circle: epitope bin A antibodies (Ab1 and Ab4). Green circle: epitope bin B antibodies (Ab2 and Ab5). Orange circle: Epitope bin C antibodies (Ab3, Ab6, and Ab7). (Note 2) A red, double-headed arrow indicates antibody pairing with strong detection capability. A yellow, double-headed arrow indicates antibody pairing with weakened detection capability. (Note 3) A black dotted line indicates the neutralization functionality; inside the dotted line represents strong neutralization activity, and being on or outside the dotted line represents weak or no neutralization activity.

**Table 1 diagnostics-11-01190-t001:** Epitope bin and pairing summary.

Detector	Epitope Bin A	Epitope Bin B	Epitope Bin C
	Ab1 & Ab4	Ab2 & Ab5	Ab3, Ab6, & Ab7
**Capture**	Ab2, Ab3, Ab5, Ab6, & Ab7	Ab1, Ab3, Ab4, Ab6, & Ab7	Ab1, Ab2, Ab4, & Ab5

**Table 2 diagnostics-11-01190-t002:** NC_50_ summary table. Semi-logarithmic curve fitting was used to extrapolate the NC_50_ values. For antibody Ab5, the range of 0.5~10 μg/mL was used for curve fitting. For the other six antibodies and the combined antibodies, the range of 0.1~10 μg/mL was used for curve fitting. (Note 1) Strong neutralizing activity: NC_50_ < 1.5 μg/mL. Moderate neutralizing activity (blue): 1.5 μg/mL ≤ NC_50_ < 15 μg/mL. Weak or no neutralizing activity (orange): NC_50_ ≥ 15 μg/mL. (Note 2) ** indicates that the NC_50_ value was not calculated due to the flat nature of the curve and the very weak neutralizing activity.

Neutralization Target	NC_50_ (μg/mL)							
	Ab1	Ab2	Ab1 + Ab2	Ab3	Ab4	Ab5	Ab6	Ab7
**SARS-CoV-2 RBD**	0.63	1.43	0.50	1.66	0.69	3.27	0.66	0.34
**UK Variant RBD**	1.08	1.35	0.82	1.51	0.95	27.63	2.72	0.86
**SA Variant RBD**	0.91	1.83	0.66	3.19	0.70	24.03	1.14	**
**BR P.1 Variant RBD**	0.70	1.40	0.46	2.21	0.69	44.79	1.11	**

## Data Availability

The data reported in this report are available from the authors upon request.

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
