# Peer review of "Use of Lateral Flow Immunoassay to Characterize SARS-CoV-2 RBD-Specific Antibodies and Their Ability to React with the UK, SA and BR P.1 Variant RBDs"

_diagnostics, 2021, doi:10.3390/diagnostics11071190_

Round 1

Reviewer 1 Report

The manuscript by Tan et al. is a thorough and well-written report describing the use of lateral flow immunoassay in characterisation of antibodies to SARS-CoV-2 and the ability of the antibodies to react with the virus receptor binding domains.

Through the use of lateral flow assays, the authors have provided information on antibody characterisation, epitope classification and antibody neutralisation kinetics as well as information concerning the effects of particular mutations in virus variants.

My only concern had been a general one of using of lateral flow devices as an alternative to other immunoassay methodologies but the others have covered this in the Discussion.

My only

Reviewer 2 Report

Tan et al in this study developed a flow dipstick assay using gold nanoparticles and recombinant SARS-CoV-2 virus RBDs to characterize seven monoclonal antibodies (mAbs) generated from SARS-CoV-2 subunits. Comprehensive pre-characterization of these antibodies will be helping moving it forward to clinical use. Specificity is one of the advantages of mAbs which can discriminate different strains or even different epitopes and variants.  Though the seven mAbs in study are not able to discriminate each of the four RBDs of COVID19 and three variants,  a tremendous work is appreciated.  Here I have some technique questions which the authors could briefly clarify.

  1. The screen/selection method for monoclonal antibodies (mAbs) during the hybridoma process directly affects what kind of binding features of the mAbs you will obtain.  The immunogens used have been described in text. Can the authors briefly tell what immune detection method was used in this selection?
  2. In line 172, can the authors describe a little about the Ab-AuNP detector? Is it a static label so the color intensity will not continuingly change after the end point or a dynamic sensitive enzyme label so a stop solution must be added after the end point? This information may give a sense how the detection color will be consistent in each test.
  3. The authors defined this in silico flow dipstick assay as neutralization assay. Strictly speaking, it was an inhibition assay while neutralization assay is a functional assay which requires cell culture or animal tests. A RBD-ACE2 binding inhibition could not necessarily mean a protection for infection.  It would be helpful if the authors could have an indication or cite a reference for this definition.
  4. The authors have tried to map the epitopes. An antigenic epitope or determinant was known to consists of five or six amino acids. Is it possible to map the 7 mAbs in the RBD sequences using short peptides?
  5. The flow dipstick assay has been developed with seven mAbs spots and two controls and has been well done for the binding epitope analysis and the four RBD analysis. The AC2 immobilized flow dipstick was also developed and tested in this study. Can these two flow dipstick assays directly be used for clinical sample tests by some kind of experiment design?
